# Effects of Polycyclic Aromatic Hydrocarbons on the Composition of the Soil Bacterial Communities in the Tidal Flat Wetlands of the Yellow River Delta of China

**DOI:** 10.3390/microorganisms12010141

**Published:** 2024-01-11

**Authors:** Yue Qi, Yuxuan Wu, Qiuying Zhi, Zhe Zhang, Yilei Zhao, Gang Fu

**Affiliations:** 1Chinese Research Academy of Environmental Sciences, Beijing 100012, China220220932760@lzu.edu.cn (Q.Z.);; 2Institute of Geographical Sciences, Heibei Academy of Sciences, Shijiazhuang 050011, China; shandongfg@126.com

**Keywords:** bacterial community, diversity, contamination, wetland ecosystems

## Abstract

Polycyclic aromatic hydrocarbons (PAHs) are pervasive organic pollutants in coastal ecosystems, especially in tidal flat wetlands. However, the mechanisms through which PAHs impact the soil bacterial communities of wetlands featuring a simple vegetation structure in the Yellow River Delta (China) remain largely unclear. In this study, we examined soil samples from two sites featuring a single vegetation type (*Suaeda salsa*) in the Yellow River Delta. Specifically, we investigated the impacts of PAHs on the diversity and composition of soil bacteria communities through high-throughput 16 S rRNA sequencing. PAHs significantly increased the soil organic carbon content but decreased the total phosphorus content (*p* = 0.02). PAH contamination notably reduced soil bacterial community α diversity (Shannon index) and β diversity. Furthermore, PAHs significantly altered the relative abundance of bacterial phyla, classes, and genera (*p* < 0.05). Specifically, PAHs increased the relative abundance of the bacterial phyla Acidobacteriota and Gemmatimonadota (*p* < 0.05), while decreasing the relative abundance of Bacteroidota, Desulfobacterota, and Firmicutes compared to the control wetland (*p* < 0.05). Moreover, PAHs and certain soil properties [total nitrogen (TN), soil organic carbon (SOC), total phosphorus (TP), and total salt (TS)] were identified as key parameters affecting the community of soil bacteria, with the abundance of specific bacteria being both negatively and positively affected by PAHs, SOC, and TN. In summary, our findings could facilitate the identification of existing environmental problems and offer insights for improving the protection and management of tidal flat wetland ecosystems in the Yellow River Delta of China.

## 1. Introduction

Soil pollution poses a significant threat to China’s environmental security. Among the most notorious soil pollutants affecting China’s ecosystems, polycyclic aromatic hydrocarbons (PAHs) represent a type of organic compound featuring a fused benzene ring structure, and are characterized by their high toxicity [1]. These compounds primarily originate from the incomplete combustion of coal, petroleum, and other fossil fuels, as well as certain biomass sources [2]. China’s annual PAH emissions have been estimated to reach up to 114,000 tons [3], with 1.4% of all examined soil sites exceeding the maximum allowable PAH concentration [4]. Individual sites contain tens of thousands of PAHs per kilogram of soil [5]. Furthermore, PAHs exert extensive effects on soil ecosystems, including causing the death and mutation of animals and plants [6,7], inhibiting plant growth [8], and engaging in complex interactions with microorganisms, influencing their diversity, composition, and physiological functions [9,10]. However, microorganisms play a key role in degrading PAHs [11,12].

Soil microbes play crucial roles in biogeochemical cycles, bioremediation, plant growth, and the preservation of soil health [13,14,15,16]. Thus, the composition and structure of microbial communities are highly sensitive to xenobiotic substances in soils [17]. Soils become contaminated with PAHs via wet or dry deposition, impacting soil microbial metabolism processes [18] and enzyme activities [19], thereby influencing various ecological processes [20].

Many studies reported that some microorganisms can degrade both PAHs and alkanes [21,22]. Moreover, many studies found that some microorganisms included the PAH and alkane degradation genes, such as benzene, toluene, ethylbenzene, xylene, and naphthalene, can be linked to similar genes, and isolated the function microorganisms in different environments [23,24]. For example, Ivanova et al. reported that Paraburkholderia aromaticivorans strain BN5 included a total of 29 monooxygenase and 54 dioxygenase genes related to the biodegradation of different hydrocarbons [21]. Therefore, the utilization of the microorganisms that degraded the PAHs has been a focus worldwide. However, to date, there is little information on the bacterial composition under PAHs contamination conditions, which inhibited the development of biodegradation microorganisms.

Numerous studies have investigated the impact of PAHs on soil microbial community structure and diversity. However, previous studies have reported contrasting results regarding the effects of PAHs in the structure and diversity of bacterial communities. Some studies have reported that PAHs exert clear adverse effects on the microbial community of soils [25]. For instance, Dai et al. [26] found that adding PAHs significantly reduced the diversity of bacterial communities in farmland soil and altered the bacterial community’s structural composition. Wu et al. [27] also reported that PAHs notably decreased the richness and diversity of soil bacterial communities in mangrove wetlands. Nevertheless, Yang et al. [28] reported contrasting results, indicating that higher levels of PAH contamination increased bacterial abundance and diversity in urban contaminated soil. Moreover, Mali et al. [29] suggested that PAHs had no significant effects on soil bacterial diversity and composition in deltaic lagoon sediments. Therefore, there is still significant uncertainty regarding the effects of varying PAH concentrations on the diversity and structure of soil microbial communities in different ecosystems.

The Yellow River Delta coastal tidal wetland constitutes a crucial component of the Bohai Sea Coastal Wetland and represents the most significant type of natural wetland in the Yellow River Delta Wetland. It serves as a vital transfer station and migration site for birds traveling inland from Northeast Asia and the Western Pacific Rim, holding substantial importance as an overwintering habitat for global biodiversity conservation. Over the past 50 years, the Yellow River Delta has evolved into a key mining area for the Shengli Oilfield and has become a pivotal oil industry base in China. The development of the petroleum industry increases the risk of discharging PAHs into the environment, thereby posing a threat to the health of wetland ecosystems. However, despite the significance of the Yellow River Delta wetland, existing research has primarily focused on the pollution characteristics and status of PAHs in inland waters and Yellow River estuary wetlands. For instance, Qi et al. [30] investigated PAHs in 120 surface sediment samples from seven sites in the Yellow River Delta coastal wetlands. The average content of 16 PAHs in the sediments was 415 ng/g, with PAHs in the 2–3 ring range being predominant, indicating a medium pollution level. However, the impact of PAH pollution in the Yellow River Delta coastal wetlands on soil bacteria diversity and community structure remains unclear. Thus, this study selected the coastal beach wetland of the Yellow River Delta as the research area, focusing on the typical Suaeda wetland in this region. Using next-generation sequencing technology, we investigated the soil bacterial community structure and quality of the Suaeda wetland contaminated by high concentrations of PAHs. Specifically, our study sought to uncover (1) whether high-concentration PAH contamination significantly alters the soil bacterial community and diversity of the Suaeda wetland, and (2) if PAHs and soil characteristics jointly lead to variations in the structure of soil bacterial communities. The findings of this study are expected to contribute valuable insights to support the ecological protection and pollution control of coastal tidal wetlands in the Yellow River Delta.

## 2. Materials and Methods

### 2.1. Site Description and Soil Sampling

Soil samples were gathered from both a PAH-contaminated wetland and a non-PAH-contaminated wetland in July 2022 (Table 1). In each wetland, we established five independent plots, each measuring 20 m × 20 m. To mitigate the influence of vegetation, careful consideration was given to maintaining the same vegetation composition and ensuring uniform vegetation height and similar coverage across all plots. Moreover, the environmental conditions (including topography, climate, and soil type) of the control plot and PAH-contaminated wetland were all the same, avoiding isolating the effects of hydrocarbons from other environmental variables. Upon removing the litter layer, ten samples were collected from each plot at a depth of 0–20 cm using a 5 cm diameter soil auger to obtain representative soil samples. Approximately 4 kg of soil was obtained from each plot. The soil samples were then promptly stored in a cooler and transported to the laboratory as quickly as possible. Next, the soil samples were combined into a single sample per plot and sieved through a 2-mm mesh to eliminate roots and other debris. The composite samples were then subdivided into two groups: one was preserved at −80 °C for microorganism analysis, and the other was allowed to air-dry for the characterization of soil physicochemical properties.

### 2.2. Physicochemical Properties of Soil

Soil pH, soil organic carbon (SOC), total nitrogen (TN), total phosphorus (TP), available nitrogen (AN), available phosphorus (AP), and PAH concentration were measured as described by [30]. Briefly, a soil–water (deionized water) (1:2.5 *w*/*v*) suspension was shaken for 30 min prior to measuring the pH with a pH meter (Thermo Scientific Orion 3-Star Benchtop, Cambridge, UK). Soil organic carbon (SOC) and the total nitrogen (TN) content were measured using an elemental analyzer (Elementar, Langenselbold, Germany). The total phosphorus (TP) content was measured using a spectrophotometer, and the available phosphorus (AP) content was measured using the colorimetric method upon extraction with 0.5 M NaHCO_3_. The available nitrogen content was measured using a continuous flow analysis system (SKALAR SAN++, Breda, The Netherlands). Three independent replicates per sample were performed for all the soil physicochemical properties. The PAH concentration was determined via gas chromatography-mass spectrometry (GC-MS) (TRACE 1300-TSQ Duo, Thermo Fisher Scientific, Waltham, MA, USA). A quartz capillary column HP-5 ms (30 m × 0.25 mm × 0.25 μm, RESTEK, Centre County, PA, USA) was equipped with helium as the carrier gas at a flow rate of 1.0 mL/min. The heating program for the chromatographic column was set at 60 °C for 2 min, heating to 110 °C at 8 °C/min for 4 min, and then heating to 240 °C at 10 °C/min, heating to 280 °C at 5 °C/min for 4 min, and then heating to 320 °C at 10 °C/min for 4 min; the detector temperature was 320 °C. The extracts were injected at 1 μL in the splitless mode. Mass spectra were acquired in the electron ionization (EI) mode and in the SIM mode using an electron impact ionization of 70 eV. The target compounds were analyzed qualitatively in the SCAN mode and quantified in the selected ion mode (SIM). Data were collected in the selected ion monitoring (SIM) of 70 eV.

### 2.3. DNA Extraction and 16 S rDNA Sequencing

To minimize deviation, six independent DNA extractions were conducted on the soil samples [31]. Approximately 500 mg of soil was utilized for each DNA extraction using the Qiagen 47016 PowerSoil Extraction Kit (Qiagen Gmbh, Hilden, Germany), following the manufacturer’s instructions. The concentration and purity of the extracted DNA were assessed using a NanoDrop ND-1000 system (Thermo Fisher Scientific, Waltham, MA, USA) and adjusted to an equal concentration of 100 ng/μL. The V3-V4 regions of the 16 S rRNA gene were amplified using the 338 F (5′- ACT CCT ACG GGA GGC AGC AG-3′) and 806 R (5′- GGA CTA CHV GGG TWT CTA AT-3′) universal primer pair [32]. The PCR amplification system consisted of a 25 μL PCR mix, 5 μL each of forward and reverse primers (5 μL/mol L^−1^), 5 μL of the DNA template, and sufficient ultrapure water (ddH_2_O) to achieve a 50 μL reaction volume. The amplification conditions consisted of an initial pre-denaturation step at 95 °C for 5 min, followed by 35 cycles of denaturation at 95 °C for 60 s, annealing at 53 °C for 60 s, and extension at 72 °C for 60 s, and a final extension step at 72 °C for 10 min. Each PCR reaction was performed in triplicate, and the PCR products were examined using 2% agarose electrophoresis. Next, the PCR amplicons were purified with the AxyPrep DNA purification kit (MAG-PCR-CL-5, Axygen Biosciences, Redwood City, CA, USA). Three independent PCR replicates were pooled in equal amounts for paired-end sequencing on the Illumina MiSeq v3 platform (2 × 300 bp). The sequencing depth was sufficient to capture the full diversity of the microbial communities in soil, including rare or low-abundance species that might play significant roles in hydrocarbon degradation (Appendix A), and the sequences were normalized according to the lowest number sequences of the sample.

### 2.4. Bioinformatics and Statistical Analysis

Paired-end clean reads were obtained using the Trimmomatic software (V0.33), thus ensuring the quality of the raw sequences. The merging of clean reads, based on the overlap between the paired-end reads, was conducted using Flash (V1.2.11). The merged reads were then clustered based on a 97% similarity threshold using the USEARCH (V10) software to generate operational taxonomic units (OTUs). Chimeric sequences and singletons were removed using the USEARCH (V10) software. These OTU sequences were then annotated against the SILVA database (version 138.1) to acquire species annotation information [33]. Non-bacterial sequences, such as those derived from mitochondria and chloroplasts, were excluded from the OTU tables before statistical analyses. Prior to conducting further analysis of α diversity, the sequences were normalized based on the lowest number of sequences for a single sample. All statistical analyses were carried out using R (V4.0.3) [34]. The variations in the soil physicochemical parameters between the two wetlands were assessed using an independent *t*-test at a significance level of 0.05 by using IBM SPSS statistics 19.0 software. All the data were normalized before statistical analysis.

## 3. Results

### 3.1. Soil Physicochemical Properties

Significant changes were observed in the concentrations of PAHs, SOC, and TP in the soil between the control (CK; i.e., non-polluted wetland) and PAH-polluted wetland (P) conditions (Table 2, *p* = 0.02), with the concentrations of PAHs and SOC in P being higher than in CK (Table 2, *p* = 0.03). In contrast, TN, TS, TP, and pH did not exhibit significant changes (Table 2, *p* = 0.15).

### 3.2. Soil Bacterial α- and β-Diversities

The CK and P experimental groups exhibited significant differences in the Shannon diversity index of the soil bacterial community (*t*-test, *p* = 0.02), whereas the Chao1, Faith_pd, and Sobs indices did not show significant differences (Table 3). PAH contamination led to a decrease in the Shannon diversity of the soil bacterial community compared to CK (Table 3). Principal coordinate analysis (PCoA) revealed a significant difference in the soil bacterial community structure between PAHs and control wetlands (PERMANOVA *p* = 0.03; Figure 1).

### 3.3. Soil Bacterial Community Composition in Two Wetlands

The relative abundances of bacterial phyla, classes, and genera exhibited significant differences between the two wetlands (Figure 2a–c and Appendix A). Here, the relative abundance only represented the relative taxonomic abundance, i.e., it did not mean biomass or the total number of bacteria. Among the 54 bacterial phyla detected across the 10 soil samples, 10 had an average abundance greater than 1% (Figure 2a). Notably, in the P wetland, the relative abundance of Gemmatimonadota and Acidobacteriota was higher than in CK, whereas the relative abundance of Desulfobacterota, Bacteroidota, and Firmicutes was lower than in CK (Appendix A).

Among the bacterial classes, with the exception of the “unclassified” category, the relative abundances of Gammaproteobacteria, Bacteroidia, and Desulfobulbia were higher in P than in CK, whereas the relative abundance of class Anaerolineae in P was lower than in CK (Appendix A). The relative abundances of Alphaproteobacteria, Actinobacteria, Thermoanaerobaculia, Desulfuromonadia, and Rhodothermia did not exhibit significant changes compared to CK (Appendix A).

Similarly, excluding unclassified genera, the relative abundances of *Woeseia*, *Dadabacteriales*, *Limibacillus*, *Halofilum*, *Rhodovibrio*, *Sulfurovum*, and *Methylophaga* were higher in P than in CK. However, compared to CK, there was no significant change in the relative abundances of *Desulfobulbus* and *Muricauda* in the P group (Appendix A).

### 3.4. Relationship between Soil Bacterial Community Composition and Soil Physicochemical Properties in Two Wetlands in the Yellow River Delta, China

Canonical correspondence analysis (CCA) revealed that soil properties, specifically PAHs, TP, SOC, TS, and TN, were the primary factors shaping the soil bacterial community (Figure 3). Particularly, the first two axes of the CCA accounted for 55.9% of the total variance. As depicted in Figure 3, PAHs, TP, SOC, TS, and TN significantly influenced the bacterial community structure (*p* = 0.03). The soil bacterial community structure at P exhibited a significant positive correlation with PAHs and TP, whereas that at CK exhibited a significant positive correlation with SOC, TN, and TS (Figure 3).

At the phylum level, the abundance of certain groups of microorganisms was closely correlated with phylum-specific soil physicochemical factors (Figure 4a). For instance, the abundance of Nitrospinota, Dadabacteria, Planctomycetota, and Acidobacteria exhibited a significant positive correlation with the content of soil PAHs and TP, whereas it was negatively correlated with the content of soil TS and SOC (Figure 4a). Firmicutes abundance exhibited a significant negative correlation with PAH content but was positively correlated with TS content (Figure 4a). Bacteroidota abundance exhibited a significant negative correlation with TP content, whereas it was positively correlated with SOC content (Figure 4a). Patescibacteria abundance exhibited a significant negative correlation with the content of soil TN (Figure 4a).

At the class level, the abundance of certain classes was also strongly correlated with class-specific soil physicochemical properties (Figure 4b). For instance, the abundance of Bacteroidia and Desulfuromonadia exhibited a significant negative correlation with PAHs and TP, but was significantly positively correlated with SOC and TS (Figure 4b). Moreover, the abundance of Rhodothermia exhibited a significant negative correlation with PAHs and TP but was significantly positively correlated with TS (Figure 4b). The abundance of Dehalococcoidia, Thermoanaerobaculia, and Dadabacteriia was significantly negatively correlated with PAHs but significantly positively correlated with SOC (Figure 4b).

Finally, genus-level analyses revealed that the abundance of certain genera was also strongly correlated with genus-specific soil physicochemical factors (Figure 4c). For example, the abundance of *Litorilinea*, *Microbulbifer*, *Hailiangium*, *Niprospina*, *Pelagibius*, *Sulfittobacter*, *Halofilum*, *Methylophaga*, *Woeseia*, *Aestuariivivens*, *Rhodovibrio*, and *Limibacillus* was significantly negatively correlated with the content of PAHs and TP but positively correlated with the SOC and TS levels (Figure 4c). The abundance of *Dadabacteriales*, *Robiginitalea*, *Nitrospina*, *Methylophaga*, and *Aestuariivivens* exhibited a significant positive correlation with the content of PAHs and TP but was negatively correlated with SOC and TS (Figure 4c). Finally, *Desulforhopalus* abundance exhibited a negative correlation with the content of soil TN (Figure 4c).

## 4. Discussion

### 4.1. Effects of PAH Pollution on Soil Bacterial Diversity

Some studies have indicated that high-concentration PAH pollution can significantly alter the diversity of soil microorganisms, community composition, and physiological activity [27,35,36]. In this study, PAH significantly decreased the content of SOC but increased the content of TP compared to CK, while other soil physicochemical properties did not change (Table 1). This indicated the diversity of the soil bacterial community dominant affected by PAH. Similarly, our findings revealed that PAH pollution reduced the Shannon diversity index of soil bacteria but did not affect the Chao1 of soil bacteria (Table 2). This could be attributed to the various forms in which PAH pollutants persist in the soil, coupled with the hydrophobic nature of PAHs. Upon binding with the soil matrix, their biological effectiveness diminishes, influencing soil bacterial composition, diversity, and abundance. Additionally, PAHs can selectively affect certain microorganisms, leading to the enrichment of degrading functional bacteria and a decrease in the abundance of sensitive bacteria [37,38]. Many PAH degradation intermediates may also possess higher biological toxicity [39]. For instance, Dai et al. [26] reported that benzoanthraquinone, an intermediate product from the degradation of BaA, is toxic to a class of ammonia-oxidizing archaea, exerting a stronger inhibitory effect. These intermediate products can also lead to the disappearance of certain species in the soil, contributing to a reduction in the Shannon index of soil bacteria. This finding aligns with the findings of [26]. However, Andreoni et al. [40] used DGGE and amplified 16 S rDNA sequences to study soil bacterial diversity in three different regions with varying concentrations of PAHs, and found that microbial diversity in different soil types was not consistent. This suggests that the changes in soil microbial diversity in response to the degree of PAH contamination are not uniform and can be influenced by soil types and other factors, which highlights the need for additional research.

However, significant changes in the Chao1 index of soil bacteria were not observed in this study, indicating that PAH pollution did not alter the number of bacterial species in the soil. The composition of soil microbial communities is not only strongly linked to soil type and nutrient content, but is also influenced by ground vegetation [41]. In this study, PAH pollution primarily altered soil SOC and TP concentrations, whereas other soil physical and chemical indicators remained unchanged. This may partially explain why the Chao1 index of soil bacteria did not change. Additionally, our study focused on the Suaeda tidal flat wetland in the Yellow River Delta, where the ground vegetation is mainly tidal flats, and its species composition and diversity have not changed significantly. This could be another reason why the Chao1 index of soil bacteria remained unchanged. Yang et al. [28] studied soil bacteria in Jinan and Hangzhou, China, under PAH pollution and found that PAHs had a significant impact on the Shannon and Chao1 indices of soil bacteria, showing spatial differences. Wu et al. [23] investigated soil bacterial diversity in various vegetation types in mangrove wetlands and also found that PAHs reduced soil bacterial diversity and abundance, with changes being inconsistent across different locations and vegetation types. Therefore, considering the diversity of PAH pollutants and the complexity of soil, further exploration is needed to understand the ecological effects of PAHs on soil microbial communities.

### 4.2. Effects of PAH Pollution on the Composition of the Soil Bacterial Community

Consistent with our first hypothesis, elevated PAH pollution caused significant changes in the soil microbial community structure of the Suaeda wetland in the Yellow River Delta (Figure 1 and Figure 2, Appendix A). PAH pollution induced alterations in soil SOC, TP, and PAH concentrations in the Suaeda wetland (Table 2), which were the primary reason for the significant changes in the composition of the soil bacterial community structure and the abundance of major bacteria. The distribution of the microbial community can fully reflect soil quality and is significantly affected by the environment, making it an effective indicator of wetland ecology. Our findings also revealed that the bacterial community structure of polluted Suaeda wetlands was mainly affected by PAHs and TP, whereas CK was primarily influenced by SOC and TS (Figure 3). This aligns with our second hypothesis. Prolonged PAH contamination increases the PAH content in the soil, potentially leading to the enrichment of degradative functional bacteria and a decrease in sensitive bacteria in the original soil [42], resulting in significant changes in the soil bacterial community structure. This finding aligns with studies by Yang et al. [28] and Andreoni et al. [40], indicating that long-term PAH pollution can significantly impact the community structure composition of soil microorganisms.

Our findings demonstrated that PAH pollution reduced the relative abundance of Bacteroidota, Desulfobacterota, and Firmicutes, while increasing the relative abundance of Acidobacteriota and Gemmatimonadota (Figure 2 and Appendix A). Wu et al. [27] reported that bacterial phyla such as Bacteroidota, Desulfobacterota, and Firmicutes were inhibited under high PAH concentrations in mangrove wetlands exposed to varying PAH concentrations, which aligns with the results of the present study. Acidobacteriota and Gemmatimonadota are oligotrophic bacteria that typically thrive in environments with poor soil conditions. Long-term PAH pollution enriches soil PAHs in wetlands, promoting the utilization and growth of oligotrophic bacteria [32]. Moreover, Wu et al. [27] and Yang et al. [28] also observed that PAH pollution can lead to a decrease in soil quality and result in significant changes in soil microbial composition. For instance, the abundances of Bacteroidetes were lowest in mangrove wetlands under PAH pollution. Su et al. [43] reported that the distribution patterns of dormant PAH-degrading bacteria induced by PAH contamination varied among different soil microbial communities. Particularly, the PAH degradation potential of these microbes increased with higher PAH concentrations [44]. This suggests that Acidobacteriota and Gemmatimonadota may have the ability to decompose PAHs and can thrive using PAHs as a carbon source. However, additional research is needed to confirm these hypotheses.

Our results only studied the changes in the diversity and structure of the soil bacterial community in PAHs pollution in the Tidal Flat Wetlands of the Yellow River Delta of China using 16 S rRNA high-throughput amplicon sequencing technology. This study provided data support for understanding the changes in the soil microorganisms under PAH contamination in temperate wetland ecosystems, and also enriched the impact of environmental pollution on the variation of microorganisms in soil. However, the method based on 16 S rRNA high-throughput amplicon sequencing technology cannot reflect the whole change in the soil bacterial community because the PCR process will lead to some bias because high-abundance microorganisms can be amplified more easily than low-abundance microorganisms. Moreover, the changes in the structure and function of soil microorganisms under long-term PAHs pollution are currently a hot research topic, and in the future, metagenomic technology should be used to study the changes in the structure and function of soil microorganisms under long-term PAHs pollution and to explain the response mechanism of soil microbial community structure and function to environmental pollution.

This study only investigated the changes in the soil bacterial community and diversity in response to PAHs contamination in the Tidal Flat Wetlands of the Yellow River Delta of China. However, the structure and composition of soil microorganisms are not only affected by PAHs, but also by other environmental factors, such as soil nutrients, vegetation composition, micro-topography, etc. In order to better reflect the impact of PAHs on the structure and diversity of soil microbial communities, it is also necessary to further increase the sample plot settings, increase the sampling density, increase the repetition of experiments, and carry out continuous monitoring for many years in the future study. In addition, soil microorganisms are significantly affected by spatiotemporal conditions, e.g., soil microbial activities were not the same in different seasons, which may lead to inconsistent changes in the impact of PAH on soil microbial structure and diversity in different seasons. Since this study only investigated the changes in soil bacterial community structure and diversity in July 2022, it is necessary to increase the changes in soil microbial structure and function in different seasons in the future, in order to understand the impact of PAH on soil microorganisms as comprehensively as possible. Moreover, this study has not yet been carried out on the function of soil microorganisms. Therefore, in the future, it is necessary to combine metagenomic technology to comprehensively carry out the impact of PAH on the structure, diversity, and function of soil microorganisms.

## 5. Conclusions

This study revealed the varying responses of the soil bacterial community to PAHs, including the abundance of operating taxonomic units, composition, and diversity in an ecosystem featuring a single vegetation type in the Tidal Flat Wetlands of the Yellow River Delta in China. PAHs induced alterations in the soil microbial communities which, in turn, resulted in unique shifts in soil organic carbon and total phosphorous in these wetlands. Notably, PAHs significantly decreased the α diversity of soil bacterial communities (Shannon index) and altered their β diversity, with Proteobacteria being the predominant bacterial phylum in both wetlands. Additionally, tidal flat wetlands exhibited the ability to degrade PAHs, particularly in riverine runoff outlets. Taken together, our findings demonstrated that soil physicochemical properties and PAHs could jointly affect the structure of soil bacterial communities. Therefore, this study offers novel insights into the impacts of PAHs on tidal flat wetlands from the perspective of soil microbial communities and their ecological function, as well as the mechanisms governing these effects.

## Figures and Tables

**Figure 1 microorganisms-12-00141-f001:**
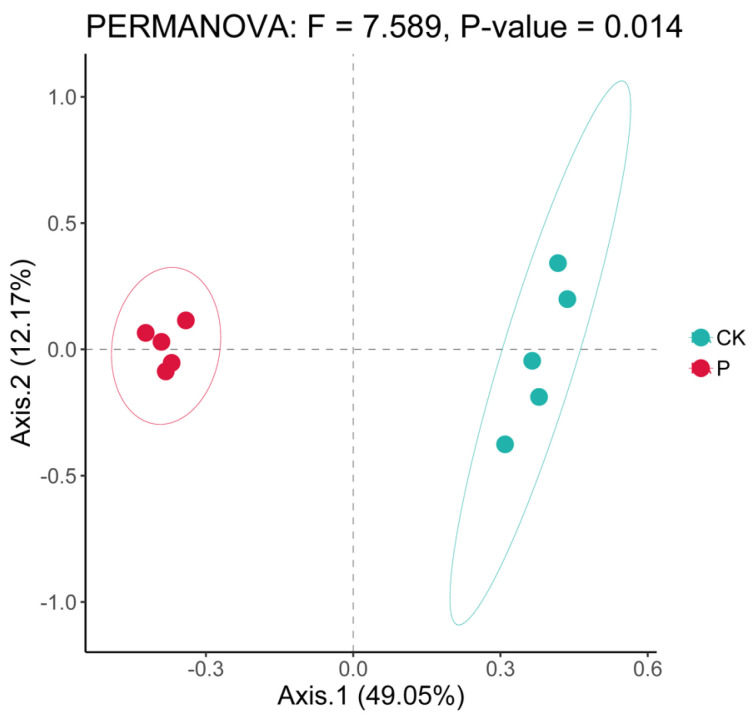
PCoA of PHA-contaminated and non-contaminated wetland soils in the Yellow River Delta, China. CK, non-polluted wetland; PAH-polluted wetland.

**Figure 2 microorganisms-12-00141-f002:**
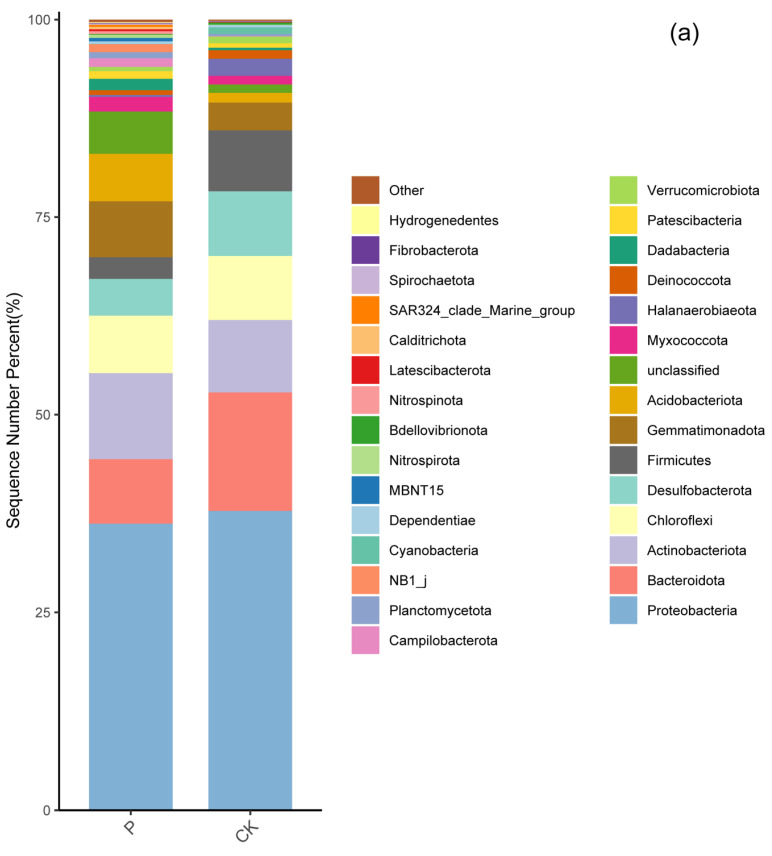
The relative abundances of bacterial operating taxonomic units (phyla, classes, and genera) exhibited significant differences between the two wetlands ((**a**–**c**) and Appendix A).

**Figure 3 microorganisms-12-00141-f003:**
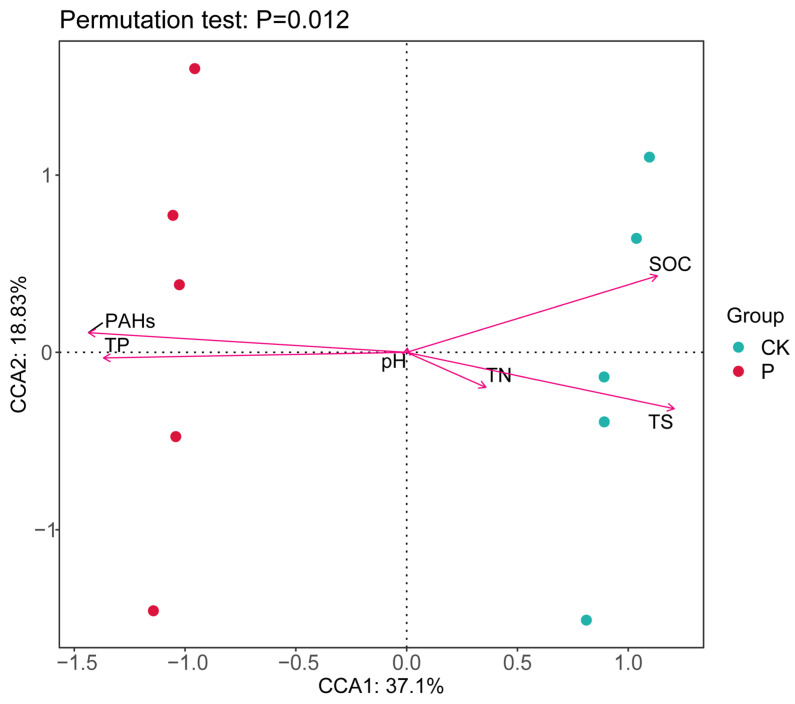
RDA of soil bacterial community structures and physicochemical properties (arrows) of PHA-contaminated and non-contaminated wetland soils in the Yellow River Delta, China.

**Figure 4 microorganisms-12-00141-f004:**
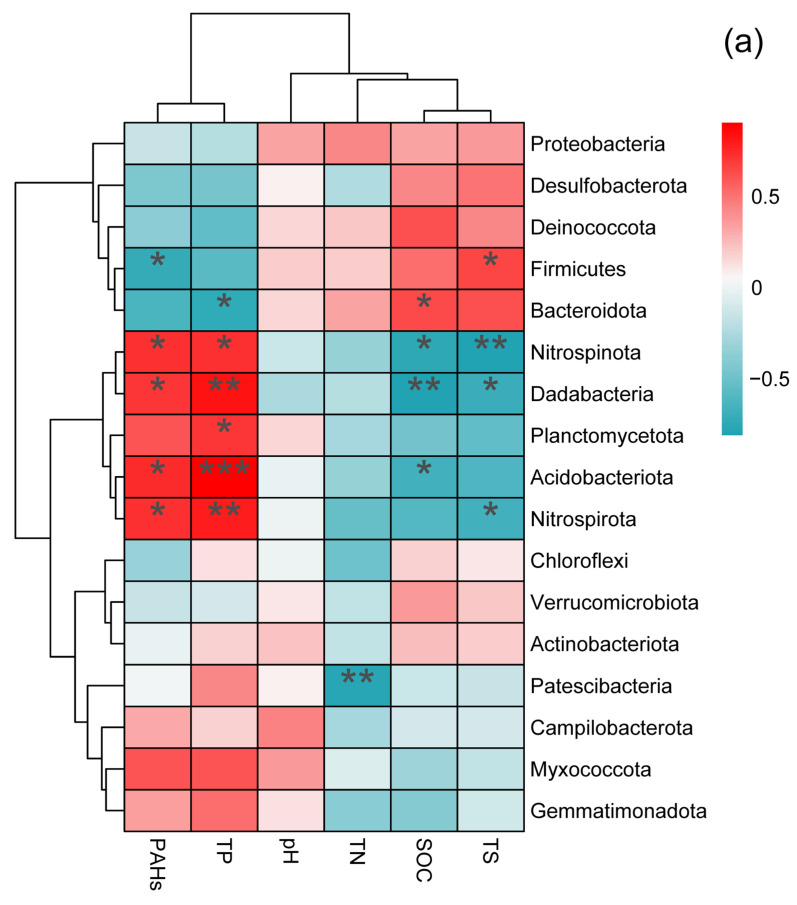
Correlation between the physicochemical properties of soil and the abundance of soil bacterial phyla (**a**), class (**b**), and genera (**c**) in PHA-contaminated and non-contaminated wetland soils in the Yellow River Delta, China. The * indicated the significant different at 0.05 level; The ** indicated the significant different at 0.01 level; The *** indicated the significant different at 0.001 level.

**Table 1 microorganisms-12-00141-t001:** Characteristics of the two wetlands (PAH-polluted and uncontaminated wetlands) in the Yellow River Delta of China.

PAHs	Latitude	Longitude	Vegetation
No pollution, (<100 ng/g, >0 ng/g)	38.12249292	118.8743	*Suaeda salsa*
Pollution, (>200 ng/g, 300 ng/g)	38.11868385	118.7122	*Suaeda salsa*

**Table 2 microorganisms-12-00141-t002:** Soil physicochemical properties of PAH-polluted and non-polluted Tidal Flat Wetlands located in the Yellow River Delta, China.

Wetlands	PAHs	SOC (g/kg)	TN (g/kg)	TS (%)	TP (g/kg)	pH
CK	46.2 ± 1.12 b	14.6 ± 1.14 b	0. 48 ± 0.01 a	0.035 ± 0.002 a	0.47 ± 0.03 a	7.27 ± 0.2 a
P	233.7 ± 4.26 a	18.1 ± 1.12 a	0. 49 ± 0.01 a	0.045 ± 0.002 a	0.26 ± 0.01 b	7.31 ± 0.2 a

CK, non-polluted wetland; P, PAH-polluted wetland; TN, total nitrogen; SOC, soil organic carbon; TS, total salt; TP, total phosphorous. Different lowercase letters indicate statistically significant differences identified through an independent *t*-test at *p* < 0.05.

**Table 3 microorganisms-12-00141-t003:** Soil bacterial α-diversity among PHA-contaminated and non-contaminated wetland soils in the Yellow River Delta, China.

Treatments	Chao1	Faith_pd	Sobs	Shannon
CK	140 ± 46.9 a	33 ± 9.9 a	140 ± 46.9 a	6 ± 0.5 a
P	142 ± 65.2 a	30 ± 13.6 a	142 ± 65.2 a	4 ± 2.1 b

Different lowercase letters indicate statistically significant differences identified through an independent *t*-test at *p* < 0.05; CK, non-polluted wetland; PAH-polluted wetland.

## Data Availability

The data presented in this study are available on request from the corresponding authors.

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
