# Peer review of "Effects of Polycyclic Aromatic Hydrocarbons on the Composition of the Soil Bacterial Communities in the Tidal Flat Wetlands of the Yellow River Delta of China"

_microorganisms, 2024, doi:10.3390/microorganisms12010141_

Round 1

Reviewer 1 Report

Comments and Suggestions for Authors

The answers to the following questions must be reflected in the "materials and methods" and in the discussion, please indicate in which sections of the document the changes will be made.

1.      How representative are the sampling sites and methods used in this study for the broader ecosystem being studied? Could additional sites or varied environmental conditions improve the representativeness of the results?

2.      Was the sequencing depth sufficient to capture the full diversity of the microbial communities, including rare or low-abundance species that might play significant roles in hydrocarbon degradation?

3.      Are the control groups and experimental designs robust enough to isolate the effects of hydrocarbons from other environmental variables? How were potential confounding factors controlled or accounted for?

4.      Does the study account for temporal variations in microbial communities? How might seasonal or other temporal factors influence the study's findings?

5.      Beyond taxonomic identification, does the study provide insights into the functional capabilities of the microbial communities regarding hydrocarbon degradation? Would integrating metagenomic or metatranscriptomic data offer a more comprehensive understanding?

6.      Are the statistical methods used in the study appropriate for the complexity of microbial community data? How were the data normalized, and what statistical tests were used to ensure robust and reliable conclusions?

7.      How do the findings of this study compare with similar ecosystems? Are the results generalizable to other environments affected by hydrocarbons?

8.      How were abiotic factors such as soil pH, temperature, and moisture content considered in the study, and what role do they play in shaping microbial communities and their response to hydrocarbon contamination?

9.      What are the key methodological limitations of the study, and how might they impact the interpretation of the results? Are there any biases inherent in the chosen methods?

10.   Based on the findings, what are the recommended directions for future research? Are there specific areas or methods that could be explored to further understand the impact of hydrocarbons on microbial communities?

Author Response

Dear Microorganisms Editorial Office,

Dear Reviewer 1 and 2 and 3,

Thank you for your letter and for the comments concerning our manuscript entitled “Effects of Polycyclic Aromatic Hydrocarbons on the Soil Bacterial Community and Diversity in Tidal Flat Wetlands of the Yellow River Delta of China (microorganisms-2750472)”. Those comments are all valuable and very helpful for revising and improving our paper. We have studied all provided comments carefully and have made appropriate corrections which we hope meet with approval. The corrections made in the paper and the respective responses to your comments are listed below and shown by revision format in the improved version of the text.

Reviewer1

The answers to the following questions must be reflected in the "materials and methods" and in the discussion, please indicate in which sections of the document the changes will be made.

  1. How representative are the sampling sites and methods used in this study for the broader ecosystem being studied? Could additional sites or varied environmental conditions improve the representativeness of the results?

Response:

Dear editor,

Thank you for your considerable comment for our manuscript. We revised our manuscript according to your comment and please see the follow and line 374-377:

“This study is only investigated the changes of soil bacterial community and diversity response to PAHs contamination in tidal flat wetlands of the yellow river delta of China. However, since the structure and composition of soil microorganisms are not only affected by PAHs, but also by other environmental factors, such as soil nutrients, vegetation composition, micro-topography, etc. In order to better reflect the impact of PAHs on the structure and diversity of soil microbial communities, It is also necessary to further increase the sample plot settings, increase the sampling density, increase the repetition of experiments, and carry out continuous monitoring for many years in the future study.”

  1. Was the sequencing depth sufficient to capture the full diversity of the microbial communities, including rare or low-abundance species that might play significant roles in hydrocarbon degradation?

Response:

Dear editor,

Thank you for your considerable comment for our manuscript. We revised our manuscript according to your comment and please see the follow and line 161-164:

“The sequencing depth was sufficient to capture the full diversity of the microbial communities in soil, including rare or low-abundance species that might play significant roles in hydrocarbon degradation (Fig. S1) and the sequences were normalized according the lowest number sequences of the sample.”.

  1. Are the control groups and experimental designs robust enough to isolate the effects of hydrocarbons from other environmental variables? How were potential confounding factors controlled or accounted for?

Response:

Dear editor,

Thank you for your considerable comment for our manuscript. We revised our manuscript according to your comment and please see the follow and line 107-109:

“… moreover, the environmental conditions (including topography, climate and soil type) of control plot and PAH-contaminated wetland were all same avoiding isolating the effects of hydrocarbons from other environmental variables.”

  1. Does the study account for temporal variations in microbial communities? How might seasonal or other temporal factors influence the study's findings?

Response:

Dear editor,

Thank you for your considerable comment for our manuscript. We revised our manuscript according to your comment and please see the follow and line 382-392:

“Another, since soil microorganisms are significantly affected by spatiotemporal conditions, e.g. soil microbial activities were not the same in different seasons, which may lead to inconsistent changes in the impact of PAH on soil microbial structure and diversity in different seasons. This study only investigated the changes in soil bacterial community structure and diversity in July 2022, it is necessary to increase the changes in soil microbial structure and function in different seasons in the future, in order to insight understand the impact of PAH on soil microorganisms as comprehensively as possible. Moreover, this study has not yet been carried out on the function of soil microorganisms. Therefore, in the future, it is necessary to combine metagenomic technology to comprehensively carry out the impact of PAH on the structure, diversity and function of soil microorganisms.”

  1. Beyond taxonomic identification, does the study provide insights into the functional capabilities of the microbial communities regarding hydrocarbon degradation? Would integrating metagenomic or metatranscriptomic data offer a more comprehensive understanding?

Response:

Dear editor,

Thank you for your considerable comment for our manuscript.

We did not analyze the function of soil bacterial community response to hydrocarbon degradation. So we are agree with your suggestion and in the next work we should investigate the function and composition combined the metagenomic technology and high-through sequence technology to systematically understand the response mechanism of soil microbiome to PAH-contamination effects.

We revised our manuscript according to your comment and please see the follow and line 389-390:

“Moreover, this study has not yet been carried out on the function of soil microorganisms. Therefore, in the future, it is necessary to combine metagenomic technology to comprehensively carry out the impact of PAH on the structure, diversity and function of soil microorganisms.”.

  1. Are the statistical methods used in the study appropriate for the complexity of microbial community data? How were the data normalized, and what statistical tests were used to ensure robust and reliable conclusions?

Response:

Dear editor,

Thank you for your considerable comment for our manuscript.

We have revised our manuscript according to your comment. Please see the follow and line 161-164:

“The sequencing depth was sufficient to capture the full diversity of the microbial communities in soil, including rare or low-abundance species that might play significant roles in hydrocarbon degradation (Fig. S1) and the sequences were normalized according the lowest number sequences of the sample.”

Line 177-179:

“The variations in soil physicochemical parameters between the two wetlands were assessed using an independent T-test at a significance level of 0.05 by using IBM SPSS statistics 19.0 software. All the data were normalized before statistical analysis.”.

  1. How do the findings of this study compare with similar ecosystems? Are the results generalizable to other environments affected by hydrocarbons?

Response:

Dear editor,

Thank you for your considerable comment for our manuscript.

We have revised our manuscript according to your comment. Please see the follow and line 284-287:

“In this study, PAH significantly decreased the content of SOC but increased the content of TP compared to CK, while other soil physicochemical properties did not changed (Table 1). This indicating the diversity of soil bacterial community dominant affected by PAH.”.

We also compared our result into other studies that performed in other ecosystems. Please see the discussion and line 316-323.

  1. How were abiotic factors such as soil pH, temperature, and moisture content considered in the study, and what role do they play in shaping microbial communities and their response to hydrocarbon contamination?

Response:

Dear editor,

Thank you for your considerable comment for our manuscript.

We have revised our manuscript according to your comment. Please see the follow and line 284-287:

“In this study, PAH significantly decreased the content of SOC but increased the content of TP compared to CK, while other soil physicochemical properties did not changed (Table 1). This indicating the diversity of soil bacterial community dominant affected by PAH.”

  1. What are the key methodological limitations of the study, and how might they impact the interpretation of the results? Are there any biases inherent in the chosen methods?

Response:

Dear editor,

Thank you for your considerable comment for our manuscript.

We have revised our manuscript according to your comment. Please see the follow and line 364-368:

“However, the method based on 16S rRNA high-throughput amplicon sequencing technology can not reflect the whole change of soil bacterial community because the PCR process will led to some bias on high abundance microorganism can be amplified easily than low abundance microorganism.”

  1. Based on the findings, what are the recommended directions for future research? Are there specific areas or methods that could be explored to further understand the impact of hydrocarbons on microbial communities?

Response:

Dear editor,

Thank you for your considerable comment for our manuscript.

We have revised our manuscript according to your comment. Please see the follow and line 359-373:

“Our results only studied the changes of diversity and structure of soil bacterial community in PAHs pollution in Tidal Flat Wetlands of the Yellow River Delta of China using 16S rRNA high-throughput amplicon sequencing technology. This study provided data support for understanding the changes of soil microorganisms under PAH con-tamination in temperate wetland ecosystems, and also enriched the impact of environ-mental pollution on the variation of microorganism in soil. However, the method based on 16S rRNA high-throughput amplicon sequencing technology can not reflect the whole change of soil bacterial community because the PCR process will led to some bias on high abundance microorganism can be amplified easily than low abundance micro-organism. Moreover, the changes in the structure and function of soil microorganisms under long-term PAHs pollution are currently a hot research topic, and in the future, metagenomic technology should be used to study the changes in the structure and func-tion of soil microorganisms under long-term PAHs pollution and to explain the response mechanism of soil microbial community structure and function to environmental pollu-tion.

This study is only investigated the changes of soil bacterial community and diver-sity response to PAHs contamination in tidal flat wetlands of the yellow river delta of China. However, since the structure and composition of soil microorganisms are not only affected by PAHs, but also by other environmental factors, such as soil nutrients, vegetation composition, micro-topography, etc. In order to better reflect the impact of PAHs on the structure and diversity of soil microbial communities, It is also necessary to further increase the sample plot settings, increase the sampling density, increase the repetition of experiments, and carry out continuous monitoring for many years in the fu-ture study. Another, since soil microorganisms are significantly affected by spatiotem-poral conditions, e.g. soil microbial activities were not the same in different seasons, which may lead to inconsistent changes in the impact of PAH on soil microbial structure and diversity in different seasons. This study only investigated the changes in soil bacte-rial community structure and diversity in July 2022, it is necessary to increase the changes in soil microbial structure and function in different seasons in the future, in or-der to insight understand the impact of PAH on soil microorganisms as comprehen-sively as possible. Moreover, this study has not yet been carried out on the function of soil microorganisms. Therefore, in the future, it is necessary to combine metagenomic technology to comprehensively carry out the impact of PAH on the structure, diversity and function of soil microorganisms.”

Reviewer 2 Report

Comments and Suggestions for Authors

This is an interesting and useful case study of the impact of polyaromatic hydrocarbon pollution on soil microbiota.

The methodology is generally appropriate, though all the conclusions about the microbiota are based upon 16S rRNA gene amplicon sequencing, with no analysis of functional genes or the wider metagenome.  This limitation needs to be specifically indicated in the Discussion section.

Line 13 (Abstract).  "Concretely" is not the right word here.  Maybe "Specifically" is meant.

Lines 51-52.  The results referred to here are not really "contradictory" because they refer to different studies where many factors may have resulted in the differing observations.  Another adjective would be better - maybe "contrasting" or "differing".

Line 220 and elsewhere.  It would be better to state actual P values rather than "< 0.05".

Comments on the Quality of English Language

Generally excellent.  Only very minor changes are needed.

Author Response

Dear Microorganisms Editorial Office,

Dear Reviewer 1 and 2 and 3,

Thank you for your letter and for the comments concerning our manuscript entitled “Effects of Polycyclic Aromatic Hydrocarbons on the Soil Bacterial Community and Diversity in Tidal Flat Wetlands of the Yellow River Delta of China (microorganisms-2750472)”. Those comments are all valuable and very helpful for revising and improving our paper. We have studied all provided comments carefully and have made appropriate corrections which we hope meet with approval. The corrections made in the paper and the respective responses to your comments are listed below and shown by revision format in the improved version of the text.

Reviewer2

This is an interesting and useful case study of the impact of polyaromatic hydrocarbon pollution on soil microbiota.

The methodology is generally appropriate, though all the conclusions about the microbiota are based upon 16S rRNA gene amplicon sequencing, with no analysis of functional genes or the wider metagenome.  This limitation needs to be specifically indicated in the Discussion section.

Response:

Thank you for your considerable comment and we have revised this point according to your comment.

Please see the line 359-373 in the discussion and follow:

“Our results only studied the changes of diversity and structure of soil bacterial community in PAHs pollution in Tidal Flat Wetlands of the Yellow River Delta of China using 16S rRNA high-throughput amplicon sequencing technology. This study provided data support for understanding the changes of soil microorganisms under PAH contamination in temperate wetland ecosystems, and also enriched the impact of environmental pollution on the variation of microorganism in soil. However, the changes in the structure and function of soil microorganisms under long-term PAHs pollution are currently a hot research topic, and in the future, metagenomic technology should be used to study the changes in the structure and function of soil microorganisms under long-term PAHs pollution and to explain the response mechanism of soil microbial community structure and function to environmental pollution.”.

Line 13 (Abstract).  "Concretely" is not the right word here.  Maybe "Specifically" is meant.

Response:

Thank you for your considerable comment and we have revised this according to your comment.

Please see the line 13 of manuscript and follow:

“Specifically, we investigated the impacts of PAHs on the diversity, composition, and function of soil bacteria communities through high-throughput 16S rRNA sequencing.”.

Lines 51-52.  The results referred to here are not really "contradictory" because they refer to different studies where many factors may have resulted in the differing observations.  Another adjective would be better - maybe "contrasting" or "differing".

Response:

Thank you for your considerable comment and we have revised this according to your comment.

Please see the line 51 of manuscript and follow:

“However, previous studies have reported contrasting results regarding the effects of PAHs in the structure and diversity of bacterial communities.”.

Line 220 and elsewhere.  It would be better to state actual P values rather than "< 0.05".

Response:

Thank you for your considerable comment and we have revised this according to your comment.

Please see the line 16, 172, 173, 174, 181 and 186 of manuscript.

Reviewer 3 Report

Comments and Suggestions for Authors

Comments and Suggestions for Authors
1. Title.

Please change the title for more precise one, otherwise, readers may also hope to find some data on the bacterial biomass or activity in this research article. I recommend the following: “Effects of Polycyclic Aromatic Hydrocarbons on Composition of the Soil Bacterial Communities in Tidal Flat Wetlands of the Yellow River Delta of China”.

2. Abstract.

2.1. Lines 13-14. As well as in the title: please eliminate word “function”and change for: “Concretely, we investigated the impacts of PAHs on the diversity and composition of soil bacteria communities through high-throughput 16S rRNA sequencing.”

2.2. Line 25. Please change “health problems” for “environmental problems”. We all can agree that environmental problems could be related with the coming health and food problems but this research article describes only wetlands and no diseases. So, I recommend: “In summary, our findings could facilitate the identification of existing environmental problems".  

3. Key words.

Please change “community” for “bacterial community”.

4. Introduction.

4.1. Lines 31-32. As well as in my comment 2.2, - it will be more correct to postulate definition “environmental security”. Links from environmental security to health (of humans, animals, etc) or food production (of rice? of fish?) can be suggested but they were not studied in this research article. So? Please change for “Soil pollution poses a significant threat to China’s environmental security.”

4.2. I also advise to add a new paragraph after the second paragraph on the page 2. The “Introduction” section has to be supplemented with some text and related references about metabolic metagenomes of the soil bacterial populations and/or genomes of strains which can provide advantages via an ability of the PAH consumption. Here I mention an example of a few fresh publications produced just by one group of authors (I’m not a co-author):

Frantsuzova, E.; Bogun, A.; Solomentsev, V.; Vetrova, A.; Streletskii, R.; Solyanikova, I.; Delegan, Y. Whole genome analysis and assessment of the metabolic potential of Gordonia rubripertincta strain 112, a degrader of aromatic and aliphatic compounds. Biology 2023, 12: 721. https://doi.org/10.3390/biology12050721

Frantsuzova, E.; Solomentsev, V.; Vetrova, A.; Travkin, V.; Solyanikova, I.; Delegan, Y. Complete genome sequence of Gordonia polyisoprenivorans 135, a promising degrader of aromatic compounds. Microbiology Resource Announcements, 2023, 12 (4): e00058-23 https://doi.org/10.1128/mra.00058-23

Ivanova, A.A.; Sazonova, O.I.; Zvonarev, A.N.; Delegan, Y.A.; Streletskii, R.A.; Shishkina, L.A.; Bogun, A.G.; Vetrova, A.A. Genome analysis and physiology of Pseudomonas sp. strain OVF7 degrading naphthalene and n-dodecane. Microorganisms. 2023, 11(8): 2058. doi: 10.3390/microorganisms11082058.

5. Materials and Methods.

Line 115. Technical misprint – please change “were” for “was”.

6. Results.

Subsection 3.3. Please here and below add a few phrases to show readers that “abundance” is only relative taxonomic abundance, i.e. it does not mean crop of biomass / total number of bacteria. Otherwise, readers can get some wrong impression (as well, in your section “Conclusions”) that you measured the PAH effects on bacterial crop or on bacterial functional activity.

For example, please change Lines 192-193 for: “The relative abundances of bacterial operating taxonomic units (phyla, classes, and genera) exhibited significant differences between the two wetlands (Figure 2a to 2c and Table S1). “

7. Discussion.

No comments.

8. Conclusions.

8.1. Lines 338-340. Once more, to make emphasis on the fact that “abundance” means only some taxonomic abundance, please, add just a few words: “This study revealed the varying responses of the soil bacterial community to PAHs, including abundance of operating taxonomic units, composition, and diversity in an ecosystem featuring a single vegetation type in tidal flat wetlands of the Yellow River Delta in China.”

Lines 340-342. Please don't reverse cause and effect. First comes pollution. Then the bacteria carry out biogeochemical processes and degrade the pollutants. As a result of these processes, the contents of organic carbon and phosphorous change in the soil. So, the more correct sentence could be: “PAHs induced alterations in the soil microbial communities which, in turn, resulted in unique shifts in soil organic carbon and total phosphorous in these wetlands.”

Author Response

Dear Microorganisms Editorial Office,

Dear Reviewer 1 and 2 and 3,

Thank you for your letter and for the comments concerning our manuscript entitled “Effects of Polycyclic Aromatic Hydrocarbons on the Soil Bacterial Community and Diversity in Tidal Flat Wetlands of the Yellow River Delta of China (microorganisms-2750472)”. Those comments are all valuable and very helpful for revising and improving our paper. We have studied all provided comments carefully and have made appropriate corrections which we hope meet with approval. The corrections made in the paper and the respective responses to your comments are listed below and shown by revision format in the improved version of the text.

Reviewer 3

Comments and Suggestions for Authors

  1. Title.

Please change the title for more precise one, otherwise, readers may also hope to find some data on the bacterial biomass or activity in this research article. I recommend the following: “Effects of Polycyclic Aromatic Hydrocarbons on Composition of the Soil Bacterial Communities in Tidal Flat Wetlands of the Yellow River Delta of China”.

Response:

Thank you for your considerable comment and we have revised this according to your comment.

Please see the line title of manuscript.

  1. Abstract.

2.1. Lines 13-14. As well as in the title: please eliminate word “function”and change for: “Concretely, we investigated the impacts of PAHs on the diversity and composition of soil bacteria communities through high-throughput 16S rRNA sequencing.”

Response:

Thank you for your considerable comment and we have revised this according to your and reviewer 2’ comment.

“Specifically, we investigated the impacts of PAHs on the diversity and composition of soil bacteria communities through high-throughput 16S rRNA sequencing.”

2.2. Line 25. Please change “health problems” for “environmental problems”. We all can agree that environmental problems could be related with the coming health and food problems but this research article describes only wetlands and no diseases. So, I recommend: “In summary, our findings could facilitate the identification of existing environmental problems". 

Response:

Thank you for your considerable comment and we have revised this according to your and comment.

  1. Key words.

Please change “community” for “bacterial community”.

Response:

Thank you for your considerable comment and we have revised this according to your and comment.

“bacterial community”

  1. Introduction.

4.1. Lines 31-32. As well as in my comment 2.2, - it will be more correct to postulate definition “environmental security”. Links from environmental security to health (of humans, animals, etc) or food production (of rice? of fish?) can be suggested but they were not studied in this research article. So? Please change for “Soil pollution poses a significant threat to China’s environmental security.”

Response:

Thank you for your considerable comment and we have revised this according to your and comment.

“Soil pollution poses a significant threat to China’s environmental security.”

4.2. I also advise to add a new paragraph after the second paragraph on the page 2. The “Introduction” section has to be supplemented with some text and related references about metabolic metagenomes of the soil bacterial populations and/or genomes of strains which can provide advantages via an ability of the PAH consumption. Here I mention an example of a few fresh publications produced just by one group of authors (I’m not a co-author):

Response:

Thank you for your considerable comment and we have revised this according to your and comment. We added a new paragraph after the second paragraph. Please see the follow:

“Many studies reported that some microorganisms can be degraded both PAHs and alkanes [21, 22]. Moreover, many studies found that some microorganisms included the PAH and alkane degradation genes such as benzene, toluene, ethylbenzene, xylene, and naphthalene, can be linked to similar genes, and isolated the function microorganisms in different environments [23, 24]. For example, Ivanova et al. reported that Paraburkholderia aromaticivorans strain BN5 included a total of 29 monooxygenase and 54 dioxygenase genes related to biodegradation of different hydrocarbons [21]. Therefore, utilization of microorganisms degraded the PAHs has been a focus across the worldwide. However, to date, there is little information on bacterial composition under PAHs contamination conditions, which inhibited the development of biodegradation microorganisms.”

Frantsuzova, E.; Bogun, A.; Solomentsev, V.; Vetrova, A.; Streletskii, R.; Solyanikova, I.; Delegan, Y. Whole genome analysis and assessment of the metabolic potential of Gordonia rubripertincta strain 112, a degrader of aromatic and aliphatic compounds. Biology 2023, 12: 721. https://doi.org/10.3390/biology12050721

Frantsuzova, E.; Solomentsev, V.; Vetrova, A.; Travkin, V.; Solyanikova, I.; Delegan, Y. Complete genome sequence of Gordonia polyisoprenivorans 135, a promising degrader of aromatic compounds. Microbiology Resource Announcements, 2023, 12 (4): e00058-23 https://doi.org/10.1128/mra.00058-23

Ivanova, A.A.; Sazonova, O.I.; Zvonarev, A.N.; Delegan, Y.A.; Streletskii, R.A.; Shishkina, L.A.; Bogun, A.G.; Vetrova, A.A. Genome analysis and physiology of Pseudomonas sp. strain OVF7 degrading naphthalene and n-dodecane. Microorganisms. 2023, 11(8): 2058. doi: 10.3390/microorganisms11082058.

  1. Materials and Methods.

Line 115. Technical misprint – please change “were” for “was”.

Response:

Thank you for your considerable comment and we have revised this according to your and comment.

  1. Results.

Subsection 3.3. Please here and below add a few phrases to show readers that “abundance” is only relative taxonomic abundance, i.e. it does not mean crop of biomass / total number of bacteria. Otherwise, readers can get some wrong impression (as well, in your section “Conclusions”) that you measured the PAH effects on bacterial crop or on bacterial functional activity.

Response:

Thank you for your considerable comment and we have revised this according to your and comment. Please see the follow and revision manuscript.

“Here the relative abundance was only represented the relative taxonomic abundance, i.e. it did not mean biomass or total number of bacteria.”.

For example, please change Lines 192-193 for: “The relative abundances of bacterial operating taxonomic units (phyla, classes, and genera) exhibited significant differences between the two wetlands (Figure 2a to 2c and Table S1). “

Response:

Thank you for your considerable comment and we have revised this according to your and comment. Please see the revision manuscript.

  1. Discussion.

No comments.

  1. Conclusions.

8.1. Lines 338-340. Once more, to make emphasis on the fact that “abundance” means only some taxonomic abundance, please, add just a few words: “This study revealed the varying responses of the soil bacterial community to PAHs, including abundance of operating taxonomic units, composition, and diversity in an ecosystem featuring a single vegetation type in tidal flat wetlands of the Yellow River Delta in China.”

Response:

Thank you for your considerable comment and we have revised this according to your and comment. Please see the revision manuscript.

“This study revealed the varying responses of the soil bacterial community to PAHs, including abundance of operating taxonomic units, composition, and diversity in an ecosystem featuring a single vegetation type in tidal flat wetlands of the Yellow River Delta in China.”

Lines 340-342. Please don't reverse cause and effect. First comes pollution. Then the bacteria carry out biogeochemical processes and degrade the pollutants. As a result of these processes, the contents of organic carbon and phosphorous change in the soil. So, the more correct sentence could be: “PAHs induced alterations in the soil microbial communities which, in turn, resulted in unique shifts in soil organic carbon and total phosphorous in these wetlands.”

Response:

Thank you for your considerable comment and we have revised this according to your and comment. Please see the revision manuscript.

“PAHs induced alterations in the soil microbial communities which, in turn, resulted in unique shifts in soil organic carbon and total phosphorous in these wetlands.”

Round 2

Reviewer 1 Report

Comments and Suggestions for Authors

The author adapted the manuscript according to the questions asked